Tissue mortality by Caribbean ciliate infection and white band disease in three reef-building coral species

Verde Alejandra averde@usb.ve 1
Bastidas Carolina 2 3
Croquer Aldo 1
1 Laboratorio de Ecología Experimental, Departamento de Estudios Ambientales, Universidad Simón Bolívar , Caracas , Venezuela
2 Laboratorio de Comunidades Marinas y Ecotoxicología, Universidad Simón Bolivar , Sartenejas , Miranda , Venezuela
3 Sea Grant College Program, Massachusetts Institute of Technology , Cambridge , MA , United States
Rodriguez-Lanetty Mauricio
Electronic publication date: 2016 Jul 26
Publication date: 2016
Volume: 4
Electronic Location ID: e2196
Received 2016 Jan 22; Accepted 2016 Jun 8
Copyright: ©2016 Verde et al.
Copyright year: 2016
Copyright holder: Verde et al.
License: This is an open access article distributed under the terms of the Creative Commons Attribution License, which permits unrestricted use, distribution, reproduction and adaptation in any medium and for any purpose provided that it is properly attributed. For attribution, the original author(s), title, publication source (PeerJ) and either DOI or URL of the article must be cited.
License URL: https://creativecommons.org/licenses/by/4.0/

Keywords: Caribbean ciliate infection, Coral diseases, White band disease, Ciliates, Corals, Venezuela

Funding: MITSUI & CO., LTD This project was funded by MITSUI & CO., LTD. The funders had no role in study design, data collection and analysis, decision to publish, or preparation of the manuscript.

==============================
Caribbean ciliate infection (CCI) and white band disease (WBD) are diseases that affect a multitude of coral hosts and are associated with rapid rates of tissue losses, thus contributing to declining coral cover in Caribbean reefs. In this study we compared tissue mortality rates associated to CCI in three species of corals with different growth forms: Orbicella faveolata (massive-boulder), O. annularis (massive-columnar) and Acropora cervicornis (branching). We also compared mortality rates in colonies of A. cervicornis bearing WBD and CCI. The study was conducted at two locations in Los Roques Archipelago National Park between April 2012 and March 2013. In A. cervicornis, the rate of tissue loss was similar between WBD (0.8 ± 1 mm/day, mean ± SD) and CCI (0.7 ± 0.9 mm/day). However, mortality rate by CCI in A. cervicornis was faster than in the massive species O. faveolata (0.5 ± 0.6 mm/day) and O. annularis (0.3 ± 0.3 mm/day). Tissue regeneration was at least fifteen times slower than the mortality rates for both diseases regardless of coral species. This is the first study providing coral tissue mortality and regeneration rates associated to CCI in colonies with massive morphologies, and it highlights the risks of further cover losses of the three most important reef-building species in the Caribbean.

Introduction

During the past few decades Caribbean coral reefs have declined partly due to the increasing prevalence of emergent and highly virulent coral diseases (Goreau et al., 1998; Harvell et al., 1999; Richardson & Aronson, 2000). Coral diseases, defined as a transitory or permanent alteration of the host physiology (Sutherland, Porter & Torres, 2004), have been often associated with bacteria (Garrett, & Ducklow, 1975; Ritchie & Smith, 1995; Richardson, 1998), fungi (Le Champion-Alsumard, Golubic & Priess, 1995; Morrison-Gardiner, 2001; Ravindran, Raghukumar & Raghukumar, 2001) or consortia of different microorganisms (Ducklow & Mitchell, 1979; Richardson, 1996). However, fewer diseases have been associated with protozoan infections (Antonius & Lipscomb, 2000; Cróquer et al., 2006).

Among protozoan infections, brown band (BB), skeletal eroding band (SEB) and Caribbean ciliate infections (CCI) are the ones with wider geographical distribution; the first two affecting a myriad of Indo-Pacific coral hosts (Page & Willis, 2008) and the latter affecting more than 25 out of the approximately 60 scleractinian species in the Caribbean (Cróquer et al., 2006). Based on microscopic examination, Rodríguez et al. (2009) suggested the name Caribbean ciliate infections (CCI) for describing Halofolliculina on Caribbean corals (Weil & Hooten, 2008; Rodríguez et al., 2009). Here, ciliate infections by Halofolliculina were first reported in 10 coral species from Venezuela (Cróquer, Bastidas & Lipscomb, 2006) but soon after that it was observed throughout the wider Caribbean (Cróquer et al., 2006). Among affected corals, Acropora palmata, A. cervicornis, Diploria labyrinthiformis, D. strigosa, Colpophyllia natans, Orbicella faveolata, O. annularis, M. franksi, Agaricia tenuifolia and Porites porites, appeared particularly vulnerable to Halofolliculina infections (Cróquer, Bastidas & Lipscomb, 2006; Page et al., 2015). In Venezuela, CCI mostly affects species of Acropora and Orbicella, reaching a prevalence of up to 85% of colonies of A. cervicornis in Los Roques (Cróquer, Bastidas & Lipscomb, 2006; Rodríguez et al., 2009). The relatively recent discovery of CCI in the Caribbean, despite disease surveys dating back to the 1970s, suggested that either the disease has recently emerged or it has been overlooked or confounded with Black Band Disease (Cróquer, Bastidas & Lipscomb, 2006; Page et al., 2015).

Experimental studies in the Caribbean demonstrated that Halofolliculina spp. transmits directly and horizontally from infected to susceptible host (Rodríguez et al., 2009). Also, the presence of lesions in corals facilitates the colonization by folliculinid ciliates (Rodríguez et al., 2009). Thus, it has been suggested that Halofolliculina infections in the Caribbean and in the Indo-Pacific (CCI and SEB, respectively) are opportunistic since they are more likely to invade damaged tissues. Aggregations of folliculinid ciliates forming scattered or dense clusters are often found in corals affected by WPD and WBD. However, factors involved in the formation of pathogenic aggregations of Halofolliculina species in CCI remain poorly understood. Seasonal environmental changes seem to affect the rate of tissue mortality of infected hosts. For instance, Rodríguez (2008) found differences in the rate of tissue mortality of CCI in Acropora palmata and A. cervicornis, being significantly higher between July and December when temperature and wind speed are higher. Moreover, Hernández (2009) found a positive and significant correlation between the rate of tissue mortality of CCI and the concentration of suspended solids in the coral A. cervicornis.

Recent studies also show that ciliates are common organisms thriving in lesions produced by other coral diseases including WBD, and whether they are scavengers or pathogens in corals with white syndromes remain controversial (Sweet & Bythell, 2012; Randall, Jordán-Garza & Van Woesik, 2015; Sweet & Séré, 2015). White band disease was first noticed in the early 80’s (Gladfelter, 1982) and was the first coral disease to cause widespread mass mortality (Gladfelter, 1982; Green & Bruckner, 2000). Multiple bacteria have been associated as the primary cause of WBD infections: (a) Ritchie & Smith (1998) and Gil-Agudelo, Smith & Weil (2006) identified Vibrio harveyi as the putative pathogen of WBD; (b) Sweet, Cróquer & Bythell (2014) identified three bacteria V. harveyi, Lactobacillus suebicus and Bacillus sp. as possible putative pathogens; and (c) Gignoux-Wolfsohn & Vollmer (2015) proposed various strains of Flavobacteriales as a new causative pathogen of WBD, although it is unknown if WBD is caused by a single or a consortium of bacteria. WBD has only been found to affect acroporid corals in the Caribbean, and two types of WBD have been described based on short-term observations of specific features of lesions (Ritchie & Smith, 1998; Bythell, Pantos & Richardson, 2004). Likely, WBD is one of the most detrimental diseases on Caribbean coral reef ecosystems as it has decimated populations of the reef building corals Acropora palmata and A. cervicornis to critical levels (Goreau et al., 1998; Richardson, 1998; Richardson & Aronson, 2000) and the presence and rapid spreading of CCI could be aggravating this plight by producing further tissue loss and hampering recovery of populations.

Figure 1 Study sites Dos Mosquises Sur and Cayo de Agua.

Map provided by Francoise Cavada and Laboratorio de Sensores Remotos.

For CCI, no studies have compared the rate of tissue mortality among coral host with different morphologies, growth forms and life strategies under natural conditions. The ability to recover and/or to heal CCI injuries is also poorly understood. In this study we estimated the rate of tissue mortality by CCI in two massive and one branching Caribbean coral species in the field (i.e., Orbicella faveolata, O. annularis and Acropora cervicornis) and their rate of tissue regeneration. We also compared the rates of tissue mortality associated to CCI and WBD in A. cervicornis.

Material and Methods

Study site

Los Roques National Park (LRNP) is an oceanic archipelago located 160 km north of the Venezuelan coast (11°44′26″–11°58′36″N, 66°32′42′–66°57′26″W; Fig. 1). The reef system encompasses more than 50 coralline cays with fringing reefs, patch reefs, over 200 sand banks, and extensive mangrove forests and seagrass beds (Weil, 2003). The study was conducted in two sites: Dos Mosquises Sur and Cayo de Agua (Fig. 1).

Estimation of mortality rates of WBD and CCI

A total of 109 coral colonies of Acropora cervicornis, Orbicella faveolata and O. annularis were tagged and observed during four field trips: April, May and November 2012 and March 2013 (Table 1). According to Work & Aeby (2006) guidelines, CCI described lesions of black color distributed diffusely with cluster of ciliates forming irregular shapes, undulating margins, distinct edges, located in the colony center or periphery for massive morphologies and in the middle for Acropora cervicornis. WBD described lesions of white color distributed diffusely with linear shapes, undulating margins, distinct edges, located in the colony base or middle. Estimations of mortality and regeneration were obtained from two sets of independent observations (April–May 2012 and November 2012–March 2013). The first set of colonies showing the classic signs of WBD (only A. cervicornis) and CCI (all three species) were tagged in April 2012 and measured in May 2012. The second group of colonies were tagged in November 2012 and measured on March 2013. Only colonies with clear and dense ciliate aggregations were selected as active CCI. Because ciliates can only be observed when they are clustered, we used a magnifying glass in the field to confirm the absence of Halofolliculina sp. in WBD corals.

Table 1 Sample sizes of coral species for each site at each sampling period.

Species/Disease	April–May 2012	November 2012–March 2013	
	Cayo de Agua	Dos Mosquises Sur	Cayo de Agua	Dos Mosquises Sur	
Orbicella annularis/CCI	9	8	6	5	
Orbicella faveolata/CCI	7	8	4	6	
Acropora cervicornis/CCI	8	7	4	6	
Acropora cervicornis/WBD	9	8	6	8	

Each coral colony was identified using aluminum tags with three stamped digits hammered with nails into dead areas in the case of massive corals and with t-raps for branching Acropora. Each colony was photographed at the start and at the end of an observation period (April–May 2012 or November 2012–March 2013). Linear rates of tissue mortality were calculated from these pictures, and for each picture a metric scale was used to convert pixels to mm.

Pictures were analyzed using the software GIMP 2.8. For this, we calculated the distance between living tissue and a reference point at each sampling time. When the difference between distances in a time period (April vs. May 2012 or November 2012 vs. March 2013) was positive, the disease had caused mortality (Figs. 2 and 3). When this difference was negative, the disease had arrested and the coral had recovered tissue from the infection (Fig. 4). Because lesions may progress in different directions, particularly in corals with massive morphologies, three measures were taken for each colony: (1) the distance from the reference point to the location of living tissues at a perpendicular angle, (2) 2.5 cm to the right and (3) 2.5 cm to the left. CCI progression occurred regardless of the position where measurements were taken (Factor “Position,” Table 2); therefore, we won’t refer to this factor hereafter.

Figure 2 Tissue mortality of Acropora cervicornis with CCI on April 2012 (A) and May 2012 (B) and with WBD (C and D, respectively).

Table 2 Univariate PERMANOVA based on Euclidean distance for the rate of tissue mortality on Acropora cervicornis, Orbicella faveolata and Orbicella annularis during April–May 2012 and November 2012–March 2013.

Bold indicates significant source of variation.

Source of variation	df	MS	F	p-value	Coefficient of variation (%)	
Rate of tissue mortality of CCI and WBD in Acropora cervicornis at Cayo de Agua and Dos Mosquises Sur	
April–May 2012						
Location	1	0.0016	0.1452	0.692	0	
Disease	1	0.0005	0.005	1	0	
Location* Disease	1	0.1085	9.597	0.009	51	
Residual	28	0.0113			49	
Total	31					
November 2012–March 2013						
Location	1	0.0014	8.2323	0.317	3.956	
Disease	1	0.0098	58.172	0.177	3.127	
Location* Disease	1	0.0002	0.0664	0.782	0	
Residual	20	0.0025			92.916	
Total	23					
Rate of tissue mortality of CCI in O. annularis and O. faveolata at Cayo de Agua and Dos Mosquises Sur	
April–May 2012						
Location	1	21.041	25.387	0.323	1.169	
Species	1	215.73	51.397	0.001	12.31	
Position (Species)	4	3.409	23.614	0.214	0.34	
Location*Species	1	0.829	0.5304	0.514	0	
Location*Position (Species)	4	0.144	461.29	0.999	0	
Residual	84	31.304			86.182	
Total	95					
November 2012–March 2013						
Location	1	0.0001	3.4718	0.505	3.109	
Species	1	0.0019	24.332	0.003	71.203	
Position (Species)	4	0.00005	1.4618	0.369	1.7	
Location*Species	1	0.00003	0.999	0.365	0	
Location*Position (Species)	4	0.00003	0.157	0.958	0	
Residual	51	0.0002			23.988	
Total	62					
Rates of tissue regeneration of CCI in Acropora cervicornis, Orbicella annularis and O. faveolata	
Species	2	0.00003	4.252	0.027	29.29	
Residual	24	0.00001			70.71	
Total	26					

Figure 3 Tissue mortality of Orbicella annularis with CCI on April 2012 (A), May 2012 (B) and November 2012 (C) and of Orbicella faveolata (D, E and F, respectively).

Figure 4 Tissue regeneration of Acropora cervicornis with WBD on April 2012 (A) and November 2012 (B).

Statistical analyses

The null hypothesis of no difference in the rate of tissue mortality produced by WBD and CCI among coral species was tested using a permutation-based analysis of variance based on Euclidean distances (PERMANOVA, Anderson, 2001). We choose PERMANOVA because of lack of normal distribution and variance homogeneity of the data. This analysis has been shown to be more robust to the violation of normality and variance homogeneity assumptions compared to other tests (Anderson & Walsh, 2013). For the data analysis we used a two factor design for Acropora cervicornis: (1) Location (random) with two levels (Cayo de Agua and Dos Mosquises Sur), (2) Disease (fixed and orthogonal to Location) with two levels (WBD and CCI). For the Orbicella species analysis we used a three factor design: (1) Location (random) with two levels (Cayo de Agua and Dos Mosquises Sur), (2) Species (fixed and orthogonal to Location) with two levels (O. annularis and O. faveolata) and (3) Position (random, nested within Species) with three levels (1, 2 and 3 concerning the three measurements made on each lesion). The analyses were performed with Primer + Permanova V. 6.1.

Results

Comparison of CCI and WBD in Acropora cervicornis

Tissue mortality of Acropora cervicornis differed significantly between diseases from April to May 2012 but showed opposite trends between sites (Fig. 5, Table 2). Colonies with WBD in Cayo de Agua lost their tissues three-fold faster than colonies with CCI (1.5 ± 1.6 mm/day, n = 9 versus 0.5 ± 0.4 mm/day, n = 8). The opposite occurred in Dos Mosquises Sur, where mortality in corals with CCI was seven-fold faster compared to colonies with WBD (1.5 ± 1.3 mm/day, n = 7 versus 0.2 ± 0.2 mm/day, n = 8) (Fig. 5, Table 2). Between November 2012 and March 2013 the rate of tissue mortality in colonies with WBD was similar to that of colonies with CCI at both sites (Fig. 5). For this period (Nov 2012–March 2013), however, there were no significant differences in mortality rates between diseases or between sites for a given disease (Table 2).

Figure 5 Rate of tissue mortality (mm/day ± SD) of CCI and WBD in Acropora cervicornis at Cayo de Agua and Dos Mosquises Sur during April–May 2012 and November 2012–March 2013.

Different letters indicate statistical significance (p < 0.05) after pairwise comparissions.

Comparison of CCI mortality between Orbicella faveolata and O. annularis

Orbicella faveolata was more vulnerable to the presence of CCI as the rate of tissue loss was 0.8 to 3-fold faster than in O. annularis. This result was consistent at both reef sites and during the two sampling periods (Fig. 6, Table 2).

Rates of recovery from CCI and WBD lesions

The tissue regeneration rate of CCI lesions was significantly different between species (Fig. 7, Table 2). Acropora cervicornis regeneration and mortality rate were higher compared to Orbicella annularis; while Orbicella faveolata regeneration and mortality rates were intermediate among species (Fig. 7).

Figure 6 Rate of tissue mortality (mm/day ± SD) of CCI in O. annularis and O. faveolata at Cayo de Agua and Dos Mosquises Sur during April–May 2012 and November 2012–March 2013.

Figure 7 Rates of tissue regeneration (mm/day ± SD) of CCI and WBD in Acropora cervicornis, Orbicella annularis and O. faveolata.

The different letters indicates significant differences.

The state of colonies at the second field trip of each sampling period was described. In March 2013 CCI was inactive in all Orbicellids colonies as well as in May 2012 for O. faveolata (Figs. 8A and 8B). In May 2012 31% of O. annularis colonies were dead (Fig. 8A). In CCI tagged A. cervicornis colonies 13% had the disease active, 40% inactive and 47% of the colonies dead on May 2012 and 80% inactive, 20% of the colonies dead on March 2013 (Fig. 8C). In WBD tagged A. cervicornis colonies 29 and 36% were dead, May 2012 and March 2013 respectively (Fig. 8D).

Figure 8 State of the colonies at the second field trip of each sampling period for O. annularis (A), O. faveolata (B), A. cervicornis with CCI (C) and A. cervicornis with WBD (D).

Oan: O. annularis, Ofav: O. faveolata, Acer: A. cervicornis. A: active, I: inactive, D: dead.

Discussion

This study provides the first estimation of mortality rates in massive Caribbean coral species with Halofolliculina infection (CCI). The results showed that CCI is more virulent in Orbicella faveolata than in O. annularis; this pattern being consistent between sites and periods of observation. Our results also indicated that CCI and White Band Disease may cause similar rates of tissue mortality in Acropora cervicornis, which is of concern as WBD is considered highly virulent. We are reporting extremely rapid rates of mortality for both diseases and concluded they are a serious threat for coral reef health.

Among species, tissue mortality in the presence of CCI was at least 2.5-fold faster in branching Acropora cervicornis than in the two massive coral species Orbicella faveolata and O. annularis. Halofolliculina ciliate infections show variations in rates of tissue loss, particularly among Caribbean coral species, supporting that taxa vary in their susceptibility as suggested by Page et al. (2015). This idea is supported by observations showing that CCI is more prevalent in species of certain genera (e.g., Diploria and Orbicella) than others (Cróquer & Weil, 2009). This is further supported with results from this study, where tissue mortality by CCI was strikingly different in two species of the genus Orbicella: tissue mortality in Orbicella faveolata was at least two-fold faster than in O. annularis. This could be due to the level of integration of the colony (i.e., the continuous connection between colony polyps) as the boulder type of O. faveolata seems more integrated than the columnar type of growth of O. annularis. This could also be related to our results of 100% of O. faveolata colonies found without CCI for the second field trip in both study periods while in May 2012 31% of O. annularis colonies were dead. It is expected that for colonies with similar living area, a higher level of tissue integration would allow more translocation of resources among polyps (and probably more efficiently too), thus affecting the mortality rate of the colony. Also, it is hypothesized that the coral colony can activate a defense mechanism more rapidly with a higher level of integration (Henry & Hart, 2005). In addition to coral morphology, pathogen virulence and host resistance to diseases also depend on intrinsic mechanisms of defense and a suite of immune responses which may be more or less efficient among coral species (Sutherland, Porter & Torres, 2004; Cróquer & Weil, 2009).

Caribbean acroporids are highly susceptible to disease epizootics, particularly to WBD which reduced their population number to critical levels on a regional scale (Goreau et al., 1998; Richardson, 1998; Richardson & Aronson, 2000). Our study supports a high mortality of Acropora cervicornis to WBD, and it also showed that this species tends to be equally affected by WBD and CCI. Furthermore, in our study most colonies were found without any disease signs after 1–3 months, which might cause an underestimation of colony mortality. Despite this, the results of this study support extremely rapid rates of progression for CCI and WBD, further suggesting that they are a serious threat for coral reef health. In addition, we observed high variability in tissue mortality of WBD and CCI in A. cervicornis, particularly associated to time of the year and site. Such natural variations might result from differential susceptibility among hosts with different genotypes or with seasonal variations. For instance, Vollmer & Palumbi (2007) reported that A. cervicornis genotypes can be more or less resistant to WBD in Panamá. In Los Roques, the rate of tissue mortality of WBD in A. cervicornis was 1.5-50-fold lower than reported values in Florida (Williams & Miller, 2005; Smith & Thomas, 2008). These results support that different populations of A. cervicornis might be more vulnerable to WBD than others depending on genetic differences and/or local environmental settings.

Environmental changes may enhance virulence in detriment of the host or promote host resistance (Weil & Cróquer, 2009). Sedimentation, eutrophication, pollution and extreme temperatures have been related with at least ten diseases (Sutherland, Porter & Torres, 2004). There is evidence that CCI also responds to environmental changes. For example, Rodríguez (2008) reported that rates of disease progression in Acropora palmata were higher in summer (August-December) (0.9 ± 0.5 mm/day) than in winter (January–May) (0.4 ± 0.5 mm/day). Further experimental evidence supports that temperature plays an important role in determining the impacts of Halofolliculina infections on corals. For instance, Rodríguez et al. (2009) demonstrated that rates of ciliate colonization on experimentally injured corals maintained at 30 °C (90% of colonies) were significantly higher compared with corals maintained at 26 °C (70% of colonies). In our study, mortality rates associated to CCI were also higher in the months with higher temperatures April–May 2012 (with a mean temperature of 27.6 °C) than in the months of lower temperatures November 2012–March 2013 (with a mean temperature of 27.2 °C), although the influence of other seasonal variables cannot be discarded.

Disease progression and tissue mortality associated with aggregations of Halofolliculina have been documented for four Indo-Pacific and three Caribbean coral species (Page & Willis, 2008; Haapkylä et al., 2009; Rodríguez, 2008; Rodríguez et al., 2009; Page et al., 2015). In the Pacific, Acropora muricata and A. pulchra had rates of disease progression of 2 ± 0.3 mm/day and 5 mm/day, respectively (Haapkylä et al., 2009). These mortality rates were three to seven-fold higher than rates obtained in Caribbean species so far: (a) this study (Acropora cervicornis: 0.7 ± 0.2 mm/day), (b) previous studies of Acropora (Acropora palmata: 0.51 ± 0.20 mm/day and Acropora cervicornis: 0.33 ± 0.18 mm/day, Rodríguez, 2008), and (c) nearly ten-fold higher than that of Agaricia tenuifolia (0.26 ± 0.08 mm/day, Rodríguez et al., 2009). These results support that Halofolliculina infection represents an important threat to the survivorship of coral reefs in the Caribbean and in the Indo-Pacific.

Our results also showed that CCI produced tissue mortality at a greater rate than Caribbean yellow band disease (CYBD) and dark spot disease (DSD), diseases that had caused significant loss of coral cover (Cróquer & Weil, 2009; Page & Willis, 2008). In addition, CCI and WBD produced tissue mortality at least ten times faster than tissue regeneration supporting the potential role that CCI could have in the loss of coral cover in the Caribbean. Tissue regeneration and repairing of wounds are complex processes which demand energy in detriment of other physiological processes such as reproduction and growth (Henry & Hart, 2005; Rodríguez et al., 2009; Weil, Cróquer & Urreiztieta, 2009). Regeneration of coral tissues, where healthy polyps cooperate with the translocation of photosynthetic products, depends on the characteristic of the lesion (i.e., size, form and position) and the level of integration of the colony (Henry & Hart, 2005; Page & Willis, 2008; Rodríguez et al., 2009). Usually, branching corals repair faster than massive ones because the modules of the former are more integrated than in the later (Henry & Hart, 2005). This notion is supported by our study, as Acropora cervicornis repaired their wounds faster than the other two massive species.

In conclusion, CCI produced differential mortality between three colony morphologies. In the branching Acropora cervicornis, it produced mortality at least 2.5 times faster than in the two massive species of Orbicella. Furthermore, in O. faveolata with a massive-boulder type of colony, tissue mortality was up to seven-fold faster than in O. annularis that has a columnar type of massive growth. We suggest that colony integration may play a role in this difference between CCI progression rates in Orbicella species, but other differences in immune response are also possible. Our study shows that tissue mortality by CCI in the two massive Orbicella species was consistent between sites and periods of observation whereas in Acropora cervicornis tissue mortality varied considerably among colonies, and between sites and diseases along time. For these three reef builders, mortality rates associated with CCI were as high as those caused by other highly virulent diseases such as WBD, WPD-II and BBD, which are capable of producing extensive losses of coral cover at a basin scale. Fastest regeneration rates were up to 15 times slower than mortality rates, further supporting that CCI is a problem of concern for coral species in the Caribbean that prompts further research and amelioration approaches.

Supplemental Information

Data S1 Raw data

Click here for additional data file.

We would like to thank Elia García and David Bone as well as the Laboratorio de Coumnidades Marinas y Ecotoxicología for providing logistic help during the field trips.

Additional Information and Declarations

Competing Interests

Author Contributions

Field Study Permissions

Data Availability

The authors declare there are no competing interests.

Alejandra Verde conceived and designed the experiments, performed the experiments, analyzed the data, wrote the paper, prepared figures and/or tables, reviewed drafts of the paper.

Carolina Bastidas contributed reagents/materials/analysis tools, wrote the paper, reviewed drafts of the paper.

Aldo Croquer conceived and designed the experiments, performed the experiments, contributed reagents/materials/analysis tools, wrote the paper, reviewed drafts of the paper.

The following information was supplied relating to field study approvals (i.e., approving body and any reference numbers):

All permits necessary to conduct this work were processed and accepted by the Governmental Venezuelan authorities (i.e., Ministerio del Poder Popular para el Ambiente-Oficina de Diversidad Biológica) and the Instituto Nacional de Parques Nacionales. PAA-123-2012.

The following information was supplied regarding data availability:

The raw data files were provided as Data S1.

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
