# Peer review of "Tissue mortality by Caribbean ciliate infection and white band disease in three reef-building coral species"

_PeerJ, doi:10.7717/peerj.2196_

## Round 0.1 · original submission · Major Revisions

· Academic Editor

Major Revisions

Some methodological aspects of the work are not clear and require further clarification. Reviewer #3 has raised questions about incongruence between the variance of the data shown in some of the figures (error bars) and the statistical results. This important concern needs explanation.

·

Basic reporting

The paper is very well written with only minor edits suggested. For ease of interpretation I have just added comment boxes to the pdf. I found it a very interesting read and a pleasure to review.

Experimental design

Detailed and interesting. A simple design but well carried out with good statistics to back up findings.

Validity of the findings

Interesting findings, the paper will/should be cited heavily. The study highlights the importance of ciliate diseases in the Caribbean and likely also the Indo-Pacific. Nice Study!!

Additional comments

The authors should be commended for writing a great, easy to read paper. Well done.

Reviewer 2 ·

Basic reporting

This article reports on two classes of Caribbean coral diseases that affect a number of coral hosts and are characterized by rapid disease progression. Specifically this study aimed to compare tissue mortality rates of 3 coral species that have very different colony morphologies: massive-boulder, massive columnar and branching. All corals are important reef building species. Authors showed that tissue mortality rates in the branching coral was similar with both CCI and white band diseases (WBD) but CCI was faster in the branching coral than in either massive corals. Authors actually measure tissue regeneration rates, which is not commonly done in disease studies, providing a nice addition to the literature. Results indicate that recovery rates are roughly 15X slower than the mortality rates in both diseases and all coral species. By linking recovery rates to disease progression rate, the authors offer valuable information for scientists and resource managers to calculate ecosystems service losses in valuable coral reef ecosystems. I feel that this type of research is well warranted in the literature. A few comments below:

ABSTRACT
First Sentence is awkward: Caribbean ciliate infection (CCI) and white band disease (WBD) are diseases that affect a multitude of coral hosts and are associated with rapid rates of tissue losses; thus, contribute with declining coral cover in Caribbean reefs.

Change to: Caribbean ciliate infection (CCI) and white band disease (WBD) are diseases that affect a multitude of coral hosts and are associated with rapid rates of tissue losses, thus contributing to declining coral cover in Caribbean reefs.

INTRODUCTION:

The authors give a very nice CCI disease review. While the English is very good, I've highlighted a few awkward examples below. Authors should screen the manuscript carefully, although these minute changes will not affect overall meaning and significance.

First Paragraph, Second Sentence: Change ..."have been often associated to bacteria..." to "...have been often associated with bacteria..."

First Paragraph, Last sentence, first paragraph: change "...However, fewer diseases have been associated to protozoan infections..." to "...However, fewer diseases have been associated with protozoan infections..."

Second paragraph, first sentence: change "... and the latter more than 25 out of the approximately 60 scleractinian species..." to "... and the latter affecting more than 25 out of the approximately 60 scleractinian species..."

Experimental design

The research questions are clearly defined, the experimental survey design is robust, experiments appear to be conducted rigorously and methods are described adequately. All methods were non invasive.

Validity of the findings

Data appear robust and statistically sound. Results and conclusions are adequately stated. There are no subjective speculations included.

Reviewer 3 ·

Basic reporting

This manuscript documents a study looking at mortality rates related to two important coral diseases. The manuscript is generally well written, although there are some minor grammatical issues here and there (the first sentence of the abstract needs to be corrected for proper tense). As stated in the review guidelines, its structure is acceptable, figures are relevant, appropriately described and labelled, and the manuscript represents an appropriate unit of publication. Also, the raw data has been included. The introduction section is detailed and appropriate and the discussion section incorporates the relevant literature. My concerns with the manuscript lie in the statistical analysis and reporting of the results which I will comment on more specifically in a separate section.

Experimental design

The experimental design is fairly well laid out but it is missing some key details and it therefore not sufficient. For instance, from the methods it is not clear how many corals of each species in each time trial for each disease were used. You can glean this information from the sample sizes reported in the results, but it should be presented in the methods as well. I would recommend including a table with this data.

The experimental design should also provide a description of what characteristics they used to identify active disease lesions when in the field and I recommend that the description follow published guidelines (e.g., Work and Aeby 2006 Diseases of Aquatic Organisms vol 70: 155-160). In addition, they need to explain how they were able to confirm that the lesions were still active when they measured the progression at the second time point. This is because if the lesions were not active when they measured them the second time, the calculated mortality rates would have been underestimated. I assume that they did not check on the corals a third time, therefore a justification for assuming that the lesions were still active when measured the second time should be given.

My primary issue with this study is with the statistical design. First, why was PERMANOVA used instead of parametric statistical approaches? I assume this is because the data violated assumptions or there were missing data, but that needs to be stated in the analysis section. Also, in the methods it states that “Position” was noted for each of the lesion progression measurements, however it is only tested in a statistical model for the CCI infections on Orbicellids. Why wasn’t it tested for in the Acropora analysis or in the analysis of all CCI mortality rates in three species? This brings me to the larger question of why two separate analyses were run for CCI mortality rates in first Orbicellids and then in Orbicellids plus Acropora. Was separate data for the Orbicellids used in each analysis? It’s not clear but I don’t think so. If the same data on CCI mortality for the Orbicellids was used in both analyses, then that is inappropriate because there would be an inflation of the type II error rate because of multiple tests. Instead, one combined analysis should be performed with CCI mortality as the response variable and species, location, and position as the independent variables.

Validity of the findings

I have several important questions about the reported findings. First, in all of the data figures (5, 6, and 7) the error bars are quite large and in almost all cases overlapping. I understand that these error bars represent standard deviation (which is larger than standard error), but in some cases the error bars are overlapping so much I’m concerned that the statistical outputs do not represent the actual results. In addition, there are several places that the authors imply that there was a difference when both the figures and that statistics indicate that there is none.

This occurs in the description of the results comparing mortality rates of the two diseases in Acropora. Specifically, on Line 134 where the authors state that “rate of tissue mortality in colonies with WBD was slightly higher than in colonies…” Also in Line 135 where they indicate that “lesions of corals located in Cayo de Agua seemed to move faster…” In both of these instances the averages were higher but the error bars are overlapping and the statistics do not support any real difference. These statements should be removed because they are not statistically supported. The only statistically significant difference in this section was an interaction between location and disease. Typically, when there is an interaction post-hoc pair-wise tests should be performed among all of the groups to determine where the specific differences lie. This was not done here that I could determine.

I also have significant concern about the results reported for CCI mortality between the two Orbicellid species. My concern about the overall design notwithstanding (see my earlier comment about how this analysis should be combined with the Acropora data), I do not see how the data reported in figure 6 could possibly have resulted in a significant effect of species as reported in the table. There is huge overlap of the error bars between the two species. The lower error bar for O. faveolata in Cayo de Agua in April-May goes below zero! Not to mention that both tops of the error bars for O. annularis are above the averages for O. faveolata for the April-May 2012 period. How is that significantly different? Perhaps I’m missing something or this is a product of using PERMANOVA and there were missing data involved, but I really need an explanation of how such large overlap in error bars could possibly produce a significant result. I am not trying to imply that the statistical results presented are errors, but I am having a difficult time reconciling the large variance of the data shown in the figures with some of the statistical results.

Additional comments

The discussion section is very detailed and appropriate based on the reported results. However, I have a couple of points that should be addressed:

On Line 164 the authors use the term “level of integration of the colony.” It becomes obvious what this means later in the discussion, but it needs to be better explained here. I believe that they are referring to the continuous connection of tissue on O. faveolata compared to the separation of tissue units on O. annularis, but this should be defined. It’s also not clear why that would that affect tissue mortality rates. The authors need to clarify what the specific mechanism would be that would cause level of tissue integration to affect CCI mortality rate. For example, is it the spatial break up of the tissue?

On Line 171 it is not clear how this study “supports a high susceptibility of Acropora cervicornis to WBD, and it also showed that this species tends to be equally vulnerable to WBD and CCI.” This statement implies that some population level measurement of susceptibility and vulnerability relative to other species was made when only mortality rates were tested and disease prevalence was not recorded. I would recommend clarifying this or taking it out.

A few other minor comments:

Line 75 – Clearly WBD is an important disease but it’s debatable whether it is the most detrimental as stated here. Arguments could be made for white plague because of outbreaks of that disease after the 2005 bleaching event. I think it best to stay away from statements like this.

Line 130 – “showed” not “showing”

Line 130 – Use of “Then, “ is odd. I would recommend deleting and just beginning the sentence with “Colonies with WBD….”

Line 131-133 – If you put the actual tissue loss rates for one comparison you should put them for the other. Specifically the sentence beginning with “The opposite occurred…” should compare the actual mortality rates like the previous sentence does.

Line 204 – 205 – It’s not typical to capitalize disease names.

---

## Round 0.2 · accepted · Accept

· Academic Editor

Accept

The authors addressed satisfactorily all the comments and suggestions brought up by the reviewers.